# Removal of Paracetamol Using Effective Advanced Oxidation Processes

**DOI:** 10.3390/ijerph16030505

**Published:** 2019-02-11

**Authors:** Francesca Audino, Jorge Mario Toro Santamaria, Luis J. del Valle Mendoza, Moisès Graells, Montserrat Pérez-Moya

**Affiliations:** 1Center for Process and Environmental Engineering CEPIMA, Chemical Engineering Department, Universitat Politècnica de Catalunya, Escola d’Enginyeria de Barcelona Est (EEBE), Av. Eduard Maristany, 16, 08019 Barcelona, Spain; francesca.audino@upc.edu (F.A.); moises.graells@upc.edu (M.G.); 2Institute for Interfacial Engineering and Biotechnology, Fraunhofer, Nobelstrasse 12, 70569 Stuttgart, Germany; jorge.toro@igb.fraunhofer.de; 3Barcelona Research Center in Multiscale Science and Engineering, Chemical Engineering Department, Universitat Politècnica de Catalunya, Escola d’Enginyeria de Barcelona Est (EEBE), Av. Eduard Maristany, 16, 08019 Barcelona, Spain; luis.javier.del.valle@upc.edu

**Keywords:** photo-induced oxidation, Fenton, photo-Fenton, hydrogen peroxide dosage, kinetic model, cytotoxicity, paracetamol, by-products

## Abstract

Fenton, photo-Fenton, and photo-induced oxidation, were investigated and compared for the treatment of 0.26 mmol L^−1^ of paracetamol (PCT) in a deionised water matrix, during a reaction span of 120.0 min. Low and high Fenton reagent loads were studied. Particularly, the initial concentration of Fe^2+^ was varied between 0.09 and 0.18 mmol L^−1^ while the initial concentration of H_2_O_2_ was varied between 2.78 and 11.12 mmol L^−1^. The quantitative performance of these treatments was evaluated by: (i) measuring PCT concentration; (ii) measuring and modelling TOC conversion, as a means characterizing sample mineralization; and (iii) measuring cytotoxicity to assess the safe application of each treatment. In all cases, organic matter mineralization was always partial, but PCT concentration fell below the detection limit within 2.5 and 20.0 min. The adopted semi-empirical model revealed that photo induced oxidation is the only treatment attaining total organic matter mineralization ($\xi^{\mathrm{MAX}}=$ 100% in 200.0 min) at the expense of the lowest kinetic constant (k = 0.007 min^−1^). Conversely, photo-Fenton treatment using high Fenton reagent loads gave a compromise solution ($\xi^{\mathrm{MAX}}=$ 73% and k = 0.032 min^−1^). Finally, cytotoxicity assays proved the safe application of photo-induced oxidation and of photo-Fenton treatments using high concentrations of Fenton reagents.

## 1. Introduction

The increasing sensitivity of analytical instruments has allowed the detection of a growing number of new organic substances in wastewater, natural water and groundwater. Particularly, special attention has recently been paid to Contaminants of Emerging Concern (CECs) a group of chemicals including pharmaceuticals and personal care products that resist treatment by conventional wastewater treatment plants (WWTPs) and accordingly require alternative and more effective treatment processes.

Several studies have reported the presence of persistent pharmaceuticals in effluents from conventional WWTPs and in surface waters affected by such effluents [1,2,3,4,5]. Other studies have reported the presence of pharmaceuticals also in groundwaters, the major source of drinking waters [6].

Advanced Oxidation Processes (AOPs) have been widely investigated as a promising and effective alternative for the removal of such Contaminants of Emerging Concern (CECs).

Particularly, several studies have addressed the use of photo-Fenton process for the removal of microcontaminants [7] and the treatment of several kinds of industrial wastewaters (e.g., pharmaceuticals, olive-oil, cork, dye, pesticides wastewaters) with a high organic content [8]. The photo-Fenton process has attracted widespread attention in the scientific community because of the possibility of using solar light for its activation but also because it can be combined with conventional and cheaper biological processes to develop innovative, efficient and cost effective treatments [9].

The photo-Fenton process results from the photo-induced enhancement of the Fenton process. The Fenton process [10] occurs by means of the addition of hydrogen peroxide to ${\mathrm{Fe}}^{2+}$ salts (see Equations (1–3)) while the photo-Fenton process occurs by the additional use of UV-VIS light irradiation at wavelength higher than 300 nm [11,12], which produces the photolysis of ${\mathrm{Fe}}^{3+}$ complexes and subsequently causes a faster ${\mathrm{Fe}}^{2+}$ regeneration (see Equation (4)):(1)$${\mathrm{Fe}}^{2+}+H_{2}O_{2}\to {\mathrm{Fe}}^{3+}+{\mathrm{HO}}^{-}+{\mathrm{HO}}^{•}$$
(2)$${\mathrm{Fe}}^{3+}+H_{2}O_{2}↔{\mathrm{FeOOH}}^{2+}+H^{+}$$
(3)$${\mathrm{FeOOH}}^{2+}\to {\mathrm{Fe}}^{2+}+{\mathrm{HO}}_{2}{}^{•}$$
(4)$$\mathrm{Fe}{\left(\mathrm{OH}\right)}^{2+}\overset{\mathrm{hv}}{\to }{\mathrm{Fe}}^{2+}+{\mathrm{HO}}^{•}$$

The extensive experimental work dedicated in the last decades to the understanding of the photo-Fenton process has revealed the importance of the Fenton reagent ratio (H_2_O_2_:Fe^2+^) as one of the most significant factors for the enhancement of its performance [13]. 

Furthermore, several studies [14,15,16] have also shown the importance of developing an efficient hydrogen peroxide dosage strategy aimed at avoiding or limiting the activation of inefficient reactions scavenging hydrogen peroxide:(5)$$H_{2}O_{2}+{\mathrm{HO}}^{•}\to {\mathrm{HO}}_{2}{}^{•}+H_{2}O$$

Photochemical AOPs, as the photo-Fenton process, are processes that use UV/VIS radiation as a means to generate hydroxyl radicals in presence of oxidants and/or photo-catalysts. However, hydroxyl radicals can also be generated by photolysing water using the higher energies of the vacuum ultraviolet or VUV (λ < 190nm); in such a case, the process is known as VUV photo-induced oxidation.

A literature review reveals the great potential of VUV photo-induced oxidation for achieving the oxidation and mineralization of organic pollutants. Particularly, relevant studies have investigated the use of VUV photo-induced oxidation for the treatment of solvents (e.g., 1,1,1-trichloroethane) in concentrations between 1.00 and 24.90 mg L^−1^ [17]; of phenols (e.g., 2,4-diclorophenol) in concentrations in the order of 100.00 mg L^−1^ [17,18,19]; and also the treatment of organic contaminants such as 3-amino-5-methylisoxazole [20] and 1,2-dichloroethene [17] in concentrations of 49.00 and 20.00 mg L^−1^, respectively.

The photochemical reaction known as homolysis of water is shown in Equation (6): (6)$$H_{2}O\overset{\mathrm{hv}}{\to }{\mathrm{HO}}^{•}+H^{•}+H^{+}+e_{\mathrm{aq}}^{-}$$

The chemical bond dissociation leads to the formation of $H^{+},{\;H}^{•},$ hydrated electrons $e_{\mathrm{aq}}^{-}$, and the main reactive and non-selective species that is the ${\mathrm{HO}}^{•}$ radical.

The VUV photo-induced technology is an interesting alternative to other AOPs as well as a solution suitable to be combined with conventional treatments due to several advantages such as: (i) no requirement of chemical additives, (ii) completely unselective oxidation, (iii) high light-energy density, and (vi) flexibility as scalable process.

The present study investigates the performance of different AOPs (Fenton, photo-Fenton and VUV photo-induced oxidation) in the degradation of paracetamol (PCT) in aqueous solution.

The quantitative performance of the AOPs was addressed by:Measuring the evolution of PCT concentration, aimed at estimating the time at which PCT concentration decays below a threshold (HPLC limit detection),Measuring and modelling, following a semi-empirical modeling approach, the Total Organic Carbon (TOC) conversion, aimed at estimating and comparing the limits of sample mineralization,Measuring cytotoxicity at planned intervals (trading-off the effort of this analysis and the information produced), aimed at determining the real environmental outcome of the treatment.

Paracetamol was selected as model contaminant because it is one of the most frequently prescribed analgesics and antipyretics worldwide. It has been detected in concentrations up to 11.30 µg L^−1^ in European WWTPs effluents [3,5,21,22,23,24] but also in natural waters in concentrations greater than 65.00 µg L^−1^ in the Tyne River, UK [5,25].

In all cases, the initial concentration of the target compound was fixed to 0.26 mmol L^−1^ (40.00 mg L^−1^) corresponding and to an initial TOC concentration of 2.16 mmol L^−1^ (25.92 mg L^−1^), which is a value higher than that of the concentrations observed in wastewaters and surface waters [3,5,21,22,23]. This higher concentration was fixed with the aim of simulating the treatment of a real-wastewater characterized by higher PCT concentrations, as also studied by other authors. For example, a recent study [26] addressed the Fenton treatment of a paracetamol wastewater of a pharmaceutical industry characterized by a PCT concentration between 37.00 and 294.00 mg L^−1^. Finally, this value also simplifies the monitoring of PCT and TOC concentrations along the treatment span.

In all cases, deionised water was set as the water matrix in order to study the pure and specific degradation of paracetamol and its by-products, as well as their effect on the toxicity evolution. This also prevents the interference of other organic substances that are present in a real wastewater matrix with an uncertain composition. 

In the case of Fenton and photo-Fenton processes, the effects of Fenton reagents (H_2_O_2_ and Fe^2+^) and the effects of H_2_O_2_ dosage were also analyzed for further insight and discussion. Especially, low (2.78 mmol L^−1^ of H_2_O_2_ and 0.09 mmol L^−1^ of Fe^2+^ –94.50 mg L^−1^ and 5.00 mg L^−1^ respectively) and high (11.12 mmol L^−1^ of H_2_O_2_ and 0.18 mmol L^−1^ of Fe^2+^ –378.00 mg L^−1^, and 10.00 mg L^−1^ respectively) concentrations of the Fenton reactants were studied.

Besides, an additional, illustrative photo-Fenton experiment based on a dosage strategy of the oxidant was performed. This additional experiment was aimed at comparing the performance of the VUV photo-induced and the photo-Fenton process. Indeed, the dosage of the oxidant led to a continuous generation of the ${\mathrm{HO}}^{•}$ radicals as in the case of VUV photo-induced oxidation experiments. On the contrary, in the case of a photo-Fenton experiment without H_2_O_2_ dosage, the generation of ${\mathrm{HO}}^{•}$ radicals occurs mostly at the beginning of the assay.

## 2. Materials and Methods

### 2.1. Reagents and Chemicals

Paracetamol (PCT) 98% purity purchased from Sigma-Aldrich (St. Louis, MO, USA) was used as target compound. Reagent-grade hydrogen peroxide (H_2_O_2_) 33% w/v from Panreac Química SLU (Barcelona, Spain) and iron sulfate (FeSO_4_·7H_2_O) from Merck (Kenilworth, NJ, USA), adopted as the ferrous ion (Fe^2+^) source, were used as Fenton reagents. HPLC gradient grade methanol, MeOH, purchased from J.T. Baker Inc. (Phillipsburg, NJ, USA)) and filtered milli Q grade water were used as HPLC mobile phases. High purity (>99%) ascorbic acid from Riedel de Haën (Seelze, Germany)), 0.2% 1,10-phenanthroline from Scharlab SL (Barcelona, Spain), sodium acetate anhydrous and 95–98% sulfuric acid, both from Panreac Química SLU (Barcelona, Spain), were used to perform measurements of iron species. Hydrogen chloride HCl 37% from J.T. Baker Inc. (Phillipsburg, NJ, USA) was used to adjust the initial pH. Deionised water with a conductivity lower than 1.25 μs was provided by Adesco S.A. (Barcelona, Spain) and was used as water matrix in all experiments.

Dulbecco’s Modified Eagle Medium (DMEM), Phosphate-Buffered Saline (PBS) and foetal bovine serum (FBS) purchased from Gibco® (Thermo Fisher Scientific, Madrid, Spain), (3-(4,5-dimethyl-2-thiazolyl)-2,5-diphenyl-2H-tetrazolium bromide (MTT) and dimethyl sulfoxide (DMSO) reagents purchased from Sigma-Aldrich were used to perform cytotoxicity assays using cell line cultures.

### 2.2. Experimental

#### 2.2.1. Experimental Design, Analytical Determinations, Pilot Plants

The Fenton, photo-Fenton and VUV photo-induced assays were all performed in batch mode with recirculation and fixing the reaction time to 120.0 min for the treatment of a PCT water solution with an initial concentration of [PCT]^0^ = 0.26 mmol L^−1^ corresponding to an initial TOC concentration of [TOC]^0^ = 2.16 mmol L^−1^.The experiments summarized in Table 1 were set up and performed. Particularly, the design of experiments was composed of:a set of preliminary blank experiments for Fenton (dark conditions, or rather with an irradiated volume V_IRR_ = 0.0 L) and photo-Fenton (irradiated conditions, or rather with an irradiated volume V_IRR_ = 1.5 L) processes, investigating the role of the oxidant under dark (BLANK_1) and irradiated (BLANK_3) conditions, the role of the catalyst under dark (BLANK_2) and irradiated (BLANK_4) conditions, and the direct photolysis of the PCT molecule (BLANK_5);a set of Fenton experiments based on the use of low (FENTON_LOW) and high (FENTON_HIGH) concentrations of the Fenton reagents;a set of photo-Fenton experiments based on the use of low (PHOTO-FENTON_LOW) and high (PHOTO-FENTON_HIGH) concentrations of the Fenton reagents;a photo-induced oxidation experiment (VUV_PHOTO INDUCED)

The maximum initial concentration of Fe^2+^ was set to the maximum legal value in wastewaters in Spain [27]. The initial concentration of H_2_O_2_ was changed between half (low doses) and twice (high doses) the stoichiometric value (5.56 mmol L^−1^) required for the total mineralization of 0.26 mmol L^−1^ of PCT, considering H_2_O_2_ as the only oxidant in the media (Equation (7)):(7)$$C_{8}H_{9}{\mathrm{NO}}_{2}+21H_{2}O_{2}\to 8{\mathrm{CO}}_{2}+25H_{2}O+H^{+}+{\mathrm{NO}}_{3}^{-}$$

An additional illustrative photo-Fenton experiment including H_2_O_2_ dosage was also performed (coded as PHOTO-FENTON_DOSAGE).

In this case, the initial concentration of PCT was fixed to 0.26 mmol L^−1^ and the initial concentration of ferrous ion was set to 0.18 mmol L^−1^. The total amount of hydrogen peroxide to be added during the experiment (${\left[H_{2}O_{2}\right]}^{\mathrm{TOT}}$) was fixed to 11.12 mmol L^−1^ in order to evaluate the effect of a high dose of the oxidant. Particularly, an initial amount of 4.0 mL of hydrogen peroxide (2.58 mmol L^−1^) was added at the beginning of the assay. Then, 5.0 ml (3.24 mmol L^−1^) were added during the first 5.0 min (corresponding to a flowrate of 1.0 ml min^−1^) while 8.0 mL (5.18 mmol L^−1^) were added during the following 55.0 min (corresponding to a flowrate of 0.2 ml min^−1^). Hence, the total dosage time was 60.0 min.

Regarding the experimental protocol followed to perform Fenton and photo-Fenton assays, the glass reservoir was first filled with 10.0 L of deionised water. After 10.0 min of recirculation, 4.9 L of deionised water in which PCT was previously dissolved were added and were recirculated during 15.0 min with the aim of ensuring a good homogenization of the matrix. After that, a sample was taken to measure the initial concentrations of TOC and PCT ([TOC]^0^, [PCT]^0^). Once pH was adjusted to 2.8±0.2, the remaining 0.1 L of deionised water, in which Fe^2+^ was previously dissolved, were poured into the reactor and, after 10.0 min of recirculation, a sample was taken to measure the initial concentrations of the iron species ([Fe^2+^]^0^, [Fe^3+^]^0^, [Fe^TOT^]^0^). For the photo-Fenton experiments, $H_{2}O_{2}$ was added 10.0 min after the light was switched on in order to ensure the lamp to stabilize.

Contrariwise, the VUV_PHOTO-INDUCED experimental protocol consisted in filling the four tank reservoirs with 2.0-L of deionised water in which PCT had been previously dissolved (initial pH = 5.0±0.5). Then, after 15.0 min of recirculation and homogenization, a sample was taken so to measure [TOC]^0^ and [PCT]^0^, and the Xenon Excimer Flat Lamps were switched on in order to start the assay.

During Fenton and photo-Fenton experiments, pH was continuously monitored and it resulted to lay in the range pH = 2.8±0.2, which is the range defined by [28] as the range ensuring the use of the iron as a catalyst or rather, ensuring the reduction of Fe^3+^ to Fe^2+^ (Fenton-like reaction) at an appreciable rate [29] and avoiding its precipitation. Temperature was also continuously monitored and checked to lay in the range T = 28.0±2.0 °C. On the other hand, performing the photo-induced experiments required no pH adjustment; after the addition of PCT, pH naturally remained in the 5.0±0.5 range. Temperature remained at a value of 25.0±2.0 °C. 

Concentration measurements of PCT, by-products (BPs), Total Organic Carbon (TOC), H_2_O_2_ and iron species (Fe^2+^, Fe^3+^, Fe^TOT^) were carried out during the experiments. The PCT and by-products concentrations ([PCT], [BPs]) were determined using an HPLC Agilent 1200 series with UV-DAD (Agilent Technologies, Santa Clara, CA, USA) and the samples were previously treated with methanol (in proportion 50:50) in order to stop further degradation of the organic matter. The HPLC analysis used an Akady 5 μm C-18 150×4.6 mm column maintained at 25 °C as stationary phase and a mixture of methanol:water (25:75) flowing at 0.4 mL min^−1^ as mobile phase. The diode array detector was set at 243 nm.

TOC concentration ([TOC]) was measured with a VCHS/CSN TOC analyzer (Shimadzu; Kyoto, Japan). Samples were refrigerated after extraction in order to slow down any further degradation of the organic matter.

Finally, during Fenton and photo-Fenton experiments, the concentrations of hydrogen peroxide ($\left[H_{2}O_{2}\right]$) and iron species ([${\mathrm{Fe}}^{2+}$], [${\mathrm{Fe}}^{3+}$], [${\mathrm{Fe}}^{\mathrm{TOT}}$]) were determined with a U-2001 UV-VIS spectrophotometer (Hitachi, Tokyo, Japan). Standard methods [30,31] were followed to ensure the proper determination of oxidant and catalyst concentrations, respectively. Particularly, the spectrophotometric technique [30] is based on the measurement of the absorption at 450 nm of the complex formed after reaction of H_2_O_2_ with ammonium metavanadate. The 1,10-phentranoline method [31] used to analyze the evolution of iron species follows ISO 6332 and is based on the absorbance measurements of the Fe^2+^-phenantroline complex at 510 nm. Total iron concentration ($\left[{\mathrm{Fe}}^{\mathrm{TOT}}\right]$) is measured using ascorbic acid to convert all ferric ions (${\mathrm{Fe}}^{3+}$) to ferrous ions (${\mathrm{Fe}}^{2+}$); then, ferric ion concentration is determined through the iron balance ($\left[{\mathrm{Fe}}^{3+}\right]$ = $\left[{\mathrm{Fe}}^{\mathrm{TOT}}\right]$ − $\left[{\mathrm{Fe}}^{2+}\right]$).

Fenton and photo-Fenton assays were performed in a 15.0 L pilot plant composed by a glass jacketed reservoir tank and a glass annular photo-reactor (Termo Fisher Scientific, Barcelona, Spain) equipped with an Actinic BL TL-DK 36 W/10 1SL lamp (UVA-UVB) (Barcelona LED, Barcelona, Spain) with an irradiated volume of 1.5 L. A pumping system allows keeping a constant recirculation flow of 12 L min^−1^, which ensures perfect mixing. The incident photon power, E = 3.4 × 10^−4^ Einstein min^−1^ (300 and 420 nm), was measured by [32] using potassium ferrioxalate actinometry [33]. Continuous pH and temperature measurements are given by on-line sensing equipment, while a flowmeter ensures the on-line control of the recirculation flow rate. The pilot-plant is also equipped with four peristaltic pumps controlled by a PLC system connected to a SCADA system allowing reactants dosage during the experiments. For more specifications, please refer to [32,34]. Photo-induced oxidation experiments were performed in a pilot plant equipped with four Flat Lamp reactor systems, each containing a Xenon Excimer Flat Lamp emitting VUV at 172 nm and connected to a 2.0-L jacketed glassware reservoir tank. A magnetically coupled centrifugal pump Sondermann BGR 1.5 (Sondermann, Köln, Germany) is used to control the flow-rate that can be varied between 4.2 and 0.8 L min^−1^. The irradiated volume (V_IRR_) is given by the product of the lamp surface (310.0 cm^2^) for the effective absorbing path (3.6 × 10^−3^ cm) resulting in a value of 1.1 cm^3^ or rather 1.1 × 10^−3^ L. Hence, the photochemical reaction (see Equation (6)) only takes place in a volume that corresponds to the 0.056% of the total volume of 2.0 L. This pilot plant was developed at the Institute for Interfacial Engineering and Biotechnology (Fraunhofer, Germany) and for more information please refer to [35].

#### 2.2.2. Toxicity tests

The safe application of the treatments under study for the removal of PCT was also investigated. Toxicity tests based on the use of specific bacteria (*E. coli* and *S. aureus* bacteria) were performed and all the studied AOPs resulted to be environmentally friendly. The results agree the observations reported by other authors who tested the toxicity of the produced effluents on the Vibrio Fishery bacteria [36].

Hence, a system with a sensitivity higher than the one based on the use of specific bacteria was selected with the purpose of addressing a more general case study. In particular, the cytotoxicity tests based on cell lines culture were carried out. Particularly, VERO and COS-1 cells were selected and tested. Both cell lines were isolated from an African green monkey kidney [37] but their morphology is different. The morphology of VERO cells corresponds to epithelial-like cells while the morphology of COS-1 cells corresponds to fibroblast-like cells (ATCC^®^, Manassas, VA, USA).

The cells were cultured in a DMEM medium (supplemented with 100.00 U/mL penicillin, 100.00 µg mL^−1^ streptomycin and 10%-v/v FBS at 37.0 °C) and a wet atmosphere containing 5% CO_2_ and 95% air. In order to allow the formation of monolayers, VERO and COS-1 cells were seeded in culture plates of 96 wells 24 h before performing the assay. Cells with viability greater than 95% were seeded at a density of 10^4^ cells/well.

Aqueous samples of PCT were taken before, during and after the treatment by Fenton, photo-Fenton and VUV photo-induced AOP and were evaluated at serial one-third dilutions. Specifically, 150.0 µL of each PCT dilution and 150.0 µL of culture medium were added to each well in order to obtain 300.0 µL on the cell monolayer in each well. In all cases, the pH of the samples was adjusted to 7.2–7.4.

Additionally, two controls were performed: (i) a control of maximum cell growth achieved culturing the cells in medium alone (300.0 µL of culture medium) and (ii) a control to evaluate the effect of the aqueous dilution of the medium achieved by adding 150.0 µL of water and 150.0 µL of culture medium. Then, the plates were incubated for 24 h under culture conditions. The samples were evaluated in triplicate on independent plates.

Cytotoxicity was assessed by measuring the viability of the cells exposed to PCT and possible by-products. The viability was determined by the MTT method. First, after 24.0 h of culture, the media were aspirated and plates were washed with PBS (100.0 μL/well). Then, 50.0 μL of culture medium supplemented with 5.00 mg mL^−1^ MTT reagent were added to each well. The plates were incubated for 3.0 h in culture conditions to allow the formation of formazan crystals in the viable cells. The quantification was performed solubilizing the formazan crystals in a DMSO/methanol/water mixture (20/70/10%-v) using 100.0 µL/well, and the absorbance at 570 nm was measured with a microplate reader (Biochrom, Cambridge, UK).

### 2.3. Modelling

The semi-empirical model by [38] summarized in Equations (8–12), has been adopted in order to rate the progress of the degradation of organic matter through a lumped parameter such as the TOC.

The term “semi-empirical” refers to a model that represents a balanced approach between detailed first-principles modeling and pure empirical modeling (e.g., response surface). On one hand this model describes the degradation of the model contaminant by measuring a lumped parameter such as the concentration of total organic carbon. On the other hand, the model has been expressed in terms of physically meaningful factors with the aim of overcoming the limitations of pure statistical modeling.

Especially, the model describes the degradation as follows:(8)$$\frac{d\left[\mathrm{TOC}\right]}{\mathrm{dt}}=-k(\left[\mathrm{TOC}\right]-{\left[\mathrm{TOC}\right]}^{\infty })$$
where k is a kinetic constant (min^−1^) and ${\left[\mathrm{TOC}\right]}^{\infty }$ represents the limit concentration (mmol L^−1^) from which, regardless the reaction time, and under specific conditions, further degradation of the organic matter cannot be attained.

By integrating the Equation (8), the analytical expression of the TOC evolution under given initial conditions can be derived and results in:(9)$$\left[\mathrm{TOC}\right]={\left[\mathrm{TOC}\right]}^{\infty }+k({\left[\mathrm{TOC}\right]}^{0}-{\left[\mathrm{TOC}\right]}^{\infty })e^{-\mathrm{kt}}$$
where ${\left[\mathrm{TOC}\right]}^{0}$ represents the initial TOC concentration (mmol L^−1^).

TOC evolution in Equation (9) can be also expressed in terms of conversion (ξ) as follows: (10)$$\xi =\xi^{\mathrm{MAX}}\left(1-e^{-\mathrm{kt}}\right)$$
(11)$$\xi {=}^{\frac{{\left[\mathrm{TOC}\right]}^{0}-\left[\mathrm{TOC}\right]}{{\left[\mathrm{TOC}\right]}^{0}}}$$
(12)$$\xi^{\mathrm{MAX}}{=}^{\frac{{\left[\mathrm{TOC}\right]}^{0}-{\left[\mathrm{TOC}\right]}^{\infty }}{{\left[\mathrm{TOC}\right]}^{0}}}$$

Hence, this allows characterizing the process performance by determining two parameters: the maximum conversion $\xi^{\mathrm{MAX}}$ and the kinetic constant k. For more details, please refer to [38].

## 3. Results and Discussion

### 3.1. Fenton, Photo-Fenton and VUV Photo-Induced Assays

Concerning Fenton and photo-Fenton processes, results of the preliminary blank assays (not shown) revealed that, when disregarding first the catalyst and then the oxidant, PCT and TOC remained almost constant, and that the adopted UV radiation (300–420 nm) was not able to appreciably photolyze PCT.

Before starting analyzing the processes performance in terms of TOC and PCT evolution, the evolution of iron species is presented for Fenton and photo-Fenton experiments (see Figure 1). Particularly, Figure 1 shows the evolution of the concentrations of Fe^2+^, Fe^3+^ and total iron in the case of FENTON_HIGH (Figure 1a) and PHOTO-FENTON_HIGH (Figure 1b) experiments.

During the first minutes of the reaction, both cases (FENTON_HIGH and PHOTO-FENTON_HIGH) show a fast increase of Fe^3+^ concentration together with a fast decrease of Fe^2+^ concentration. In the FENTON_HIGH (Figure 1a), Fe^3+^ and Fe^2+^ concentrations reach stable values around 0.17 mmol L^−1^ and 0.01 mmol L^−1^, respectively. Conversely, the PHOTO-FENTON_HIGH case (Figure 1b) presents a different behavior. After 45.0 min, when hydrogen peroxide is depleted (see Figure 2c), the concentration of Fe^3+^ decreases while the concentration of Fe^2+^ increases up to a value of about 0.17 mmol L^−1^. Hence, Fe^2+^ becomes the main iron species in solution.

This different behavior can be explained by analyzing the Fenton and photo-Fenton kinetics together with the different hydrogen peroxide evolution. In the Fenton case, the fastest reaction is the Fenton reaction (Equation (1)), which generates Fe^3+^ from Fe^2+^ with a kinetic constant in a range of 63.0÷76.0 mol L^−1^ s^−1^ [39]. On the other hand, the only reaction generating Fe^2+^ (Equation (3)) has a kinetic constant in the range of 1.0 × 10^−2^
÷ 2.0 × 10^−2^ mol L^−1^ s^−1^ [39]. In the PHOTO-FENTON_HIGH case, unlike the FENTON_HIGH case, for which H_2_O_2_ concentration never drops to zero, the concentration of H_2_O_2_ reaches a null value. As a consequence, the Fenton reaction (Equation (1)) is no longer effective unless further Fe^2+^ is supplied (Equations (3) and (4)). The generation of Fe^2+^ is here particularly enhanced by the photo-Fenton reaction (Equation (4)) under a convenient light source (quantum yields $\bar{\Phi }$ = 0.21 ± 0.04 mol Einstein^−1^) [39]. Conversely, in the PHOTO-FENTON_DOSAGE case, the change from Fe^3+^ to Fe^2^ was not observed (results not shown). In this case, higher concentrations of Fenton reagents were used and, as a consequence, H_2_O_2_ was never depleted.

Finally, constant total iron concentration can be ensured because all the experiments were performed under a pH range (2.8±0.2) that prevents its precipitation, which is in accordance with the remarks reported by other authors [28,29] and which was experimentally confirmed as well.

Subsequently, with the aim of discussing the performance of the different treatments studied, Figure 2a–c) presents results in terms the evolution of PCT and TOC (normalized concentrations) and H_2_O_2_ (mmol L^−1^), the latter, in the case of Fenton and photo-Fenton assays, with and without H_2_O_2_ dosage.

Figure 2a allows concluding that all the treatments efficiently removed PCT. The model compound was not detected by HPLC within a minimum time of 2.5 min and a maximum time of 20.0 min. PHOTO-FENTON_LOW and PHOTO-FENTON_HIGH assays led to the best results: PCT concentration decayed beyond the HPLC detection limit in 2.5 min. Conversely, the VUV_PHOTO-INDUCED produced a slightly slower decrease of the PCT during the time, and in this case PCT concentration decayed below the HPLC detection limit in about 20.0 min.

It is worth noting that for all the studied treatments, TOC concentration after 20.0 min of reaction (maximum time at which PCT is no longer detected) was lower than that established by wastewater disposal regulations or than that required by a subsequent biological treatment. However, although all the AOPs under study allowed the efficient removal of PCT, none of them attained the complete mineralization of the organic matter within the 120.0 min reaction span.

Particularly, VUV_PHOTO-INDUCED and PHOTO-FENTON_DOSAGE experiments (Figure 2b) obtained the best results in terms of final TOC conversion for the studied time span. In both cases, a final TOC conversion of about 77% was attained, that is a 6% more if compared with the final TOC conversion reached in the case of the PHOTO-FENTON_HIGH experiment (71%). It is important to observe that such improvement was obtained also without the enhancement introduced by a convenient dosage. Hence, even better results could be expected if H_2_O_2_ dosage was approached by systematic optimization [40]. Furthermore, the VUV_PHOTO-INDUCED case also indicates that if the reaction time would had been extended (> 120.0 min), organic matter degradation would have continued, and higher levels of mineralization would have been attained. Conversely, for the PHOTO-FENTON_DOSAGE case study the TOC reached a stable value starting from a reaction time of 75.0 min so showing that an increase in reaction time would not lead to an increase in mineralization level.

This result is consistent with the hydrogen peroxide evolution observed during the experiment (Figure 2c). As a matter of fact, Figure 2c reveals that when hydrogen peroxide dosage stops (at 60.0 min), the concentration of H_2_O_2_ in the reactor quickly drops to zero, thus determining the end of the organic matter degradation.

The analysis of the experimental results solely in terms of PCT removal is not enough for sensibly deciding one AOP out of the rest. For this reason, the semi-empirical model by introduced in Section 2.3 [38] was adopted with the aim of aiding decision making. Such a model allows deriving the maximum conversion $\xi^{\mathrm{MAX}}$ and the kinetic constant k for each treatment, and conveniently display them as means to describe the process performance. Thus, Figure 3 displays the kinetic constant k_i_ as a function of the maximum attainable conversion $\xi_{i}^{\mathrm{MAX}}$, with i = VUV_PHOTO-INDUCED, FENTON_LOW, FENTON_HIGH, PHOTO-FENTON_LOW, PHOTO-FENTON_HIGH, PHOTO-FENTON_DOSAGE.

The results, as displayed in Figure 3, highlight that the photo induced advanced oxidation process is the only one that allows attaining the total organic matter mineralization ($\xi^{\mathrm{MAX}}=$ 100% in 200.0 min), but it is the process with the lowest kinetic constant (k = 0.7 × 10^−2^ min^−1^). The low rate of this process is a consequence of the insufficient illuminated volume of the pilot plant (V_IRR_ = 1.1 × 10^−3^ L), which suggests that increasing this volume would led to increasing the kinetic constant and to reaching total mineralization in less time. On the contrary, FENTON_LOW and PHOTO-FENTON_LOW show the highest kinetic constants (3.5 × 10^−2^ and 3.3 × 10^−2^ min^−1^, respectively) but only a 30% and a 41% TOC conversion, respectively.

The area highlighted in Figure 3 by a red dashed circle represents the area where the intermediate solutions are located. Particularly, the PHOTO-FENTON_HIGH and PHOTO-FENTON_DOSAGE experiments can be considered a compromise solution, since they allow reaching 73% and 84% maximum TOC conversion with a kinetic constant of 3.2 × 10^−2^ and 2.3 × 10^−2^ min^−1^, respectively.

Thus, in order to solve the trade-off between TOC conversion and kinetics, toxicity tests are considered to provide complementary results for further decision making support. This is connected with the analysis of the by-products (BPs) generated during the treatments presented in Figure 4.

For all the AOPs under study, HPLC analysis allowed detecting two main by-products that were identified as hydroquinone (HDQ) and benzoquinone (BZQ) (retention time for HDQ = 7.0 min, retention time for BZQ = 10.0 min). Since Fenton and photo-Fenton experiments produced similar results in this regard, for the sake of simplicity Figure 4 only considers photo-Fenton, which gave just a slightly faster BPs degradation than that of the Fenton case. Besides, Figure 4 only presents the evolution of BZQ in order to allow a better display of the results.

Figure 4 shows that a slightly faster decrease of BZQ was obtained by the PHOTO-FENTON_HIGH process compared to that of the FENTON_HIGH process: particularly, BZQ was no longer detected after 5.0 and 15.0 min, respectively. Contrariwise, when PHOTO-FENTON_DOSAGE assay was performed, BZQ was no longer detected after 15.0 min while, when VUV_PHOTO-INDUCED experiment was performed, BZQ concentration decayed below the HPLC detection limit in 22.0 min.

It is worth noting that in the VUV_PHOTO-INDUCED case, the ${\mathrm{HO}}^{•}$ radicals, which are responsible of the degradation of the organic matter, are formed only in the illuminated volume (1.1 × 10^−3^ L), which is also much lower than the illuminated volume of the Fenton/photo-Fenton pilot plant (1.5 L). Thus, increasing this volume would improve the kinetics of the process. Contrariwise, the presence of Fenton reaction (see Equation (1)) in the Fenton and photo-Fenton process, ensures the generation of ${\mathrm{HO}}^{•}$ radicals in all the volume of the reactor (both dark and irradiated).

It is also important to remark that the detected by-products show coherence with the literature related to the Fenton and photo-Fenton degradation of PCT [36,41] and, in addition, the analytical results obtained in this work point that the same intermediates were generated during the photo-induced AOP.

Regarding Fenton and photo-Fenton experiments, another important factor is the presence of chemical additives in high concentrations that allows a faster generation of ${\mathrm{HO}}^{•}$ radicals if compared with the VUV photo induced AOP. Actually, when hydrogen peroxide was dosed, and so when a lower concentration of H_2_O_2_ was present in the reactor at the beginning of the assay, the photo-Fenton process performance in terms of TOC, PCT, and BPs evolution, approaches the performance recorded by the photo induced AOP.

### 3.2. Cytotoxicity Assays

Even though all the AOPs under study generate hydroxyl radicals and the same by-products, the global kinetics are different and other side reactions could lead to different by-products and/or to different amounts of the same by-products and this could affect the toxicity of the samples during and after the processes. For this reason, it is important to test the toxicity of the target compound and of the treated solution during and at the end of the treatment.

Initially, the toxicity study was carried out in *E. coli* and *S. aureus* bacteria; however, both bacteria were capable of metabolizing PCT and taking advantage of it as a carbon source during its growth (data not shown). In this sense, the toxicity study has used the culture system of eukaryotic cell lines for offering greater sensitivity to PCT and its by-products to produce cellular injury.

Hence, toxicity tests based on cell lines culture were performed by using VERO and COS-1 cells (epithelial-like and fibroblast-like cells, respectively). First, the value of the LC_50_, or rather the concentration of the selected chemical that kills 50% of the tested cell in a given time, was evaluated for PCT with both line cells. Particularly, the LC_50_ of PCT resulted to be very high (>1000.0 mg L^−1^) for both line cells under study, and this result shows that PCT is tolerated by both cells. Figure 5 shows the results of the cytotoxicity assays performed to check the viability of all the treatments under study.

Particularly, results are presented for samples (M0) taken before adding the Fenton reagents (or rather before starting the VUV_PHOTO-INDUCED treatment, thus containing only PCT in a concentration of 0.26 mmol L^−1^), after 75.0 min of reaction (M75) and at the end of each treatment (M120). Cytotoxicity tests performed using samples M0 and M120 allow evaluating the toxicity of the target compound in the initial concentration set for this study and the safe application of the treatments under study for PCT removal, respectively. Moreover, the presence of possible toxic by-products was analyzed by taking samples after 75.0 min of reaction. In this way, it was possible to ensure that, in case of Fenton and photo-Fenton treatments, with the exception of FENTON_HIGH (for which a residual H_2_O_2_ concentration of about 0.38 mmol L^−1^ was detected at the end of the experiment), H_2_O_2_ is no longer present in solution. Actually, hydrogen peroxide could be toxic for the cells and may affect the toxicity tests preventing to determine if the toxicity depends on a toxic compound or the oxidant. For this reason, cytotoxicity results obtained for the FENTON_HIGH case are not shown.

Both COS-1 and VERO cells were not affected by PCT in the initial concentration of 0.26 mmol L^−1^ (see results of M0 samples in Figure 5). A strong increase of cells mortality (about 85%) was observed when both M75 and M120 samples of the FENTON_LOW and PHOTO-FENTON_LOW experiments were tested. Hence, this case reveals that it is not possible to ensure the safe application of Fenton and photo-Fenton treatments for the PCT removal for mild Fenton reagent loads. The final TOC conversion values attained by applying FENTON_LOW (31%) and PHOTO-FENTON_LOW (39%) treatments also confirm this result.

On the contrary, the PHOTO-FENTON_HIGH treatment is able to finally generate a non-toxic effluent (M120) that can be safely discharged into the aquatic environment, despite the formation of more toxic by-products during the process (highlighted by a 70% mortality of the COS-1 cells, and a 25% mortality of the VERO cells, when M75 samples were tested). A 25% mortality of the COS-1 cells, and a 10% mortality of the VERO cells was recorded when M120 samples were tested. This result is consistent with the higher mineralization value reached at the end of the treatment (71%). The slightly different result obtained using COS-1 and VERO cells can be explained by considering that both cell types have different morphological characteristics (e.g., VERO cells, as epithelial cells, have a large contact surface during cellular spreading, whereas COS-1 cells such as fibroblasts have an adhesion to the surface by focal contacts).

Finally, regarding the VUV_PHOTO-INDUCED treatment, all the cytotoxicity assays using both VERO and COS-1 cells confirm the safe application of this treatment for the removal of PCT. For both cell types, Figure 5, shows a percentage of mortality similar to the control when both M75 and M120 samples were analyzed. Moreover, despite the fact Figure 5 only shows the results for samples M0, M75 and M120, the toxicity assays for the VUV_PHOTO-INDUCED treatment were performed also using samples taken after 5.0, 15.0, 20.0, 25.0, 30.0, 45.0, 60.0, and 120.0 min. Furthermore, it is worth noting that, for all the samples, the mortality of both cell types was similar to the control. Since the intermediates detected for VUV_PHOTO-INDUCED, Fenton and photo-Fenton treatments are the same, this result can be probably due to the kinetics of the reactions, which in this case can lead to the generation of a certain amount of intermediates whose total balance determines a nontoxic solution. Alternatively, no more toxic by-products could be formed beyond those generated (even if not detected) during the Fenton and photo-Fenton processes. This can be related to the continuous generation of hydroxyl radicals during the VUV_PHOTO-INDUCED treatment; or rather, ${\mathrm{HO}}^{•}$ are not generated mostly at the beginning of the process, as it occurs in the Fenton and photo-Fenton cases instead. Hence, interesting further work could be testing the toxicity of the photo-Fenton treatment when dosing H_2_O_2_, because also in this case, as shown in the previous sections, there is a continuous generation of the hydroxyl radicals during the process.

These results allow concluding that, unlike Fenton and photo-Fenton processes, the VUV_PHOTO-INDUCED treatment can be safely stopped when it is convenient (e.g., according to the desired percentage of TOC conversion and/or PCT removal).

## 4. Conclusions

On the one hand, photo-induced oxidation resulted to be a very promising treatment. It allows the efficient removal of the model compound (which was no longer detected after 20.0 min) and it could allow the total organic matter mineralization by increasing the reaction time span to 200.0 min. Moreover, cytotoxicity assays revealed the feasibility of the VUV photo-induced treatment for PCT removal as well as the possibility to stop this process at any convenient time because it was never possible to detect any increase of toxicity.

The bottleneck of the photo-induced oxidation is the kinetics of the process, which is likely due to the illuminated volume (V_IRR_ = 1.1 × 10^−3^ L) employed, much lower than the illuminated volume of the Fenton/photo-Fenton pilot plant (V_IRR_ = 1.5 L). Therefore, further investigation is required to address the reactor design with the aim of increasing the irradiated volume. 

On the other hand, the photo-Fenton process using high concentrations of Fenton reagents demonstrated good performance in terms of both PCT removal and organic matter mineralization. However, cytotoxicity results highlighted that only after 120 min of reaction it was possible to generate a non-toxic effluent, so showing that it is not possible to stop such process at any convenient time and that is mandatory to monitor the cytotoxicity evolution.

## Figures and Tables

**Figure 1 ijerph-16-00505-f001:**
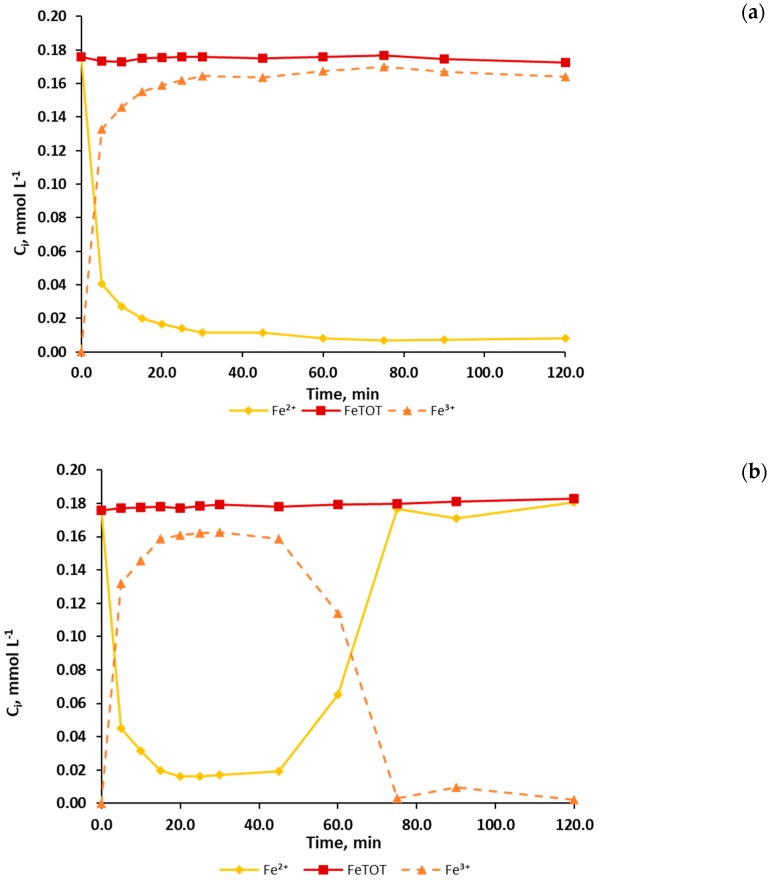
Evolution of the measured concentration of Fe^2+^ (◊,♦) and total iron (□,■) represented by a continuous line and of the calculated concentration of Fe^3+^ (∆,▲), represented by a dashed line, during (**a**) Fenton (empty symbols) and (**b**) photo-Fenton (solid symbols ) experiments performed with high concentrations of the Fenton reagents.

**Figure 2 ijerph-16-00505-f002:**
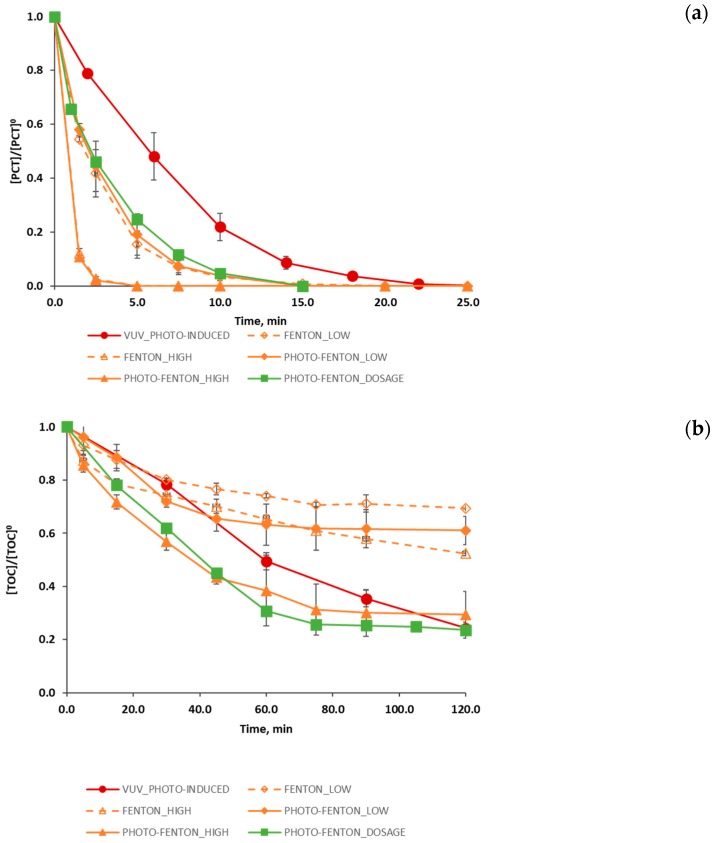
Evolution of the degradation of (**a**) PCT (normalized values), (**b**) TOC (normalized values) and (**c**) H_2_O_2_ (concentration, mmol L^−1^) for different process: photo induced advanced oxidation (●); Fenton, using low and high concentrations of Fenton reagents (◊,∆); photo-Fenton using low and high concentrations of Fenton reagents (♦,▲); and photo-Fenton process following a H_2_O_2_ dosage strategy (■). Figure 2c also shows the profile of the added hydrogen peroxide during the reaction span (concentration values, mmol L^−1^). The error bars display the standard deviation for the set of three experiments.

**Figure 3 ijerph-16-00505-f003:**
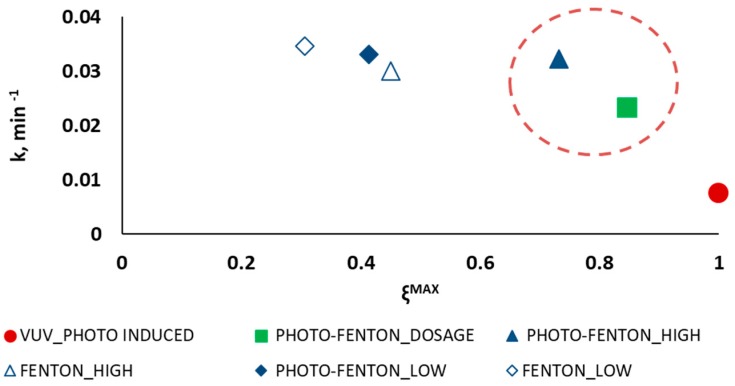
Reaction rate as a function of the maximum attainable conversion evaluated for photo induced advanced oxidation process (●); Fenton (◊) and photo-Fenton (♦) process without H_2_O_2_ dosage using low concentrations of the Fenton reagents; Fenton (∆) and photo-Fenton (▲) process without H_2_O_2_ dosage using high concentrations of the Fenton reagents; and photo-Fenton process following a H_2_O_2_ dosage strategy (■).

**Figure 4 ijerph-16-00505-f004:**
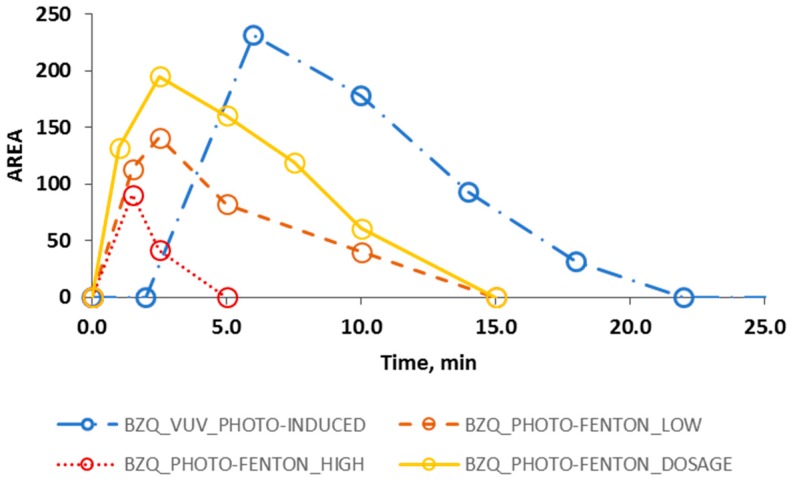
Evolution (HPLC AREA) of the degradation intermediate BZQ (○) during photo induced oxidation process (line and dots), photo-Fenton process using high (dotted line) and low (dashed line) concentrations of the Fenton reactants and without dosing the hydrogen peroxide, and photo-Fenton process following a H_2_O_2_ dosage strategy (solid line).

**Figure 5 ijerph-16-00505-f005:**
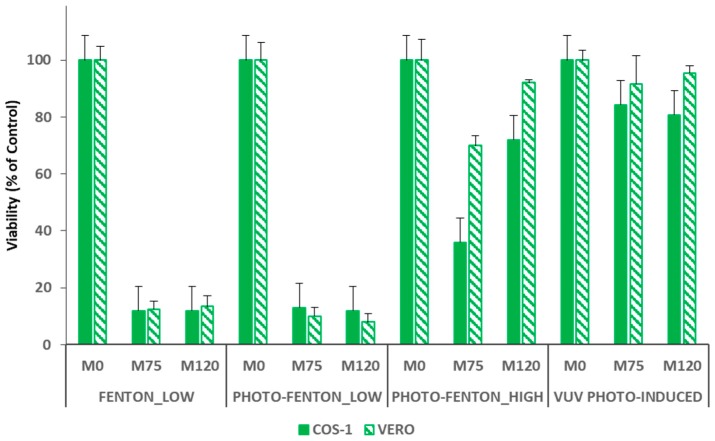
Results of the cytotoxicity assays performed using COS-1 (full symbols) and VERO (slashed symbols) and tested on samples taken after 0 (M0), 75 (M75) and 120 (M120) minutes of reaction.

**Table 1 ijerph-16-00505-t001:** Design of experiments performed for an initial PCT concentration of [PCT]^0^ = 0.26 mmol L^−1^ corresponding to an initial TOC concentration of [TOC]^0^ = 2.16 mmol L^−1^.

Experiments	[H_2_O_2_]^0^ mmol L^−1^	[Fe^2+^]^0^ mmol L^−1^	pH	T °C	λ nm	V_IRR_ L
BLANK_1	11.12	0.00	2.8 ± 0.2	28.0 ± 2	300–420	0.0
BLANK_2	0.00	0.18	2.8 ± 0.2	28.0 ± 2	300–420	0.0
BLANK_3	11.12	0.00	2.8 ± 0.2	28.0 ± 2	300–420	1.5
BLANK_4	0.00	0.18	2.8 ± 0.2	28.0 ± 2	300–420	1.5
BLANK_5	0.00	0.00	2.8 ± 0.2	28.0 ± 2	300–420	1.5
FENTON_LOW	2.78	0.09	2.8 ± 0.2	28.0 ± 2	300–420	0.0
FENTON_HIGH	11.12	0.18	2.8 ± 0.2	28.0 ± 2	300–420	0.0
PHOTO-FENTON_LOW	2.78	0.09	2.8 ± 0.2	28.0 ± 2	300–420	1.5
PHOTO-FENTON_HIGH	11.12	0.18	2.8 ± 0.2	28.0 ± 2	300–420	1.5
VUV_PHOTO INDUCED	0.00	0.00	5.0 ± 0.2	25.0 ± 2	172	1.1 × 10^−3^

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
