# Peer review of "Removal of Paracetamol Using Effective Advanced Oxidation Processes"

_ijerph, 2019, doi:10.3390/ijerph16030505_

Reviewer 1 Report

See attached file

Author Response

First of all the authors would like to thank the reviewers for the effort spent in improving with all their comments and suggestions the quality of the manuscript.

Below you can find the answers to each of the comments and questions arisen.

Reviewer #1:

v  General comment: In the present form, the article is too dense and must be mandatorily alleviated.

Response to Reviewer 1 general comment: The manuscript was alleviated by revising the materials and methods, results and discussion, and conclusions sections.

1.      Section 2, Materials and Methods, must be drastically shortened as it is unusually long. The detailed description of the experiments can be more briefly exposed. If the authors consider of importance such description, maybe they can be derived to a supplementary material document.

Response to Reviewer 1 comment No. 1: As suggested by the reviewer, the material and methods section was shortened and no supplementary material document was added.

2.      Throughout the text, please give the concentrations in mmol/L (or also in mmol/L), specially the ratios between reagents must be in terms of molar ratios as mass has no sense for reactions.

Response to Reviewer 1 comment No. 2: Throughout the text as well as in the plots of the revised manuscript, all the concentrations are now given in mmol L-1.

3.      At the start of the reaction, after mixing by recirculation, did you measured the initial PCT? Note that, given its low concentration, any adsorption over the equipment material can substantially alter the results.

Response to Reviewer 1 comment No. 3: Yes, after 15.0 min of recirculation a sample was taken in order to measure the initial concentration (at time 0.0 min) of TOC and PCT ([TOC]0, [PCT]0), in order to be sure of the starting point. This has been clarified in the revised manuscript (lines 190-191 and lines 199-200).

4.      The authors wrongly use the term reaction rate for what is a kinetic constant (or parameter), which of course still serves to indicate how favourable is the kinetics of a reaction. Please, replace throughout the manuscript.

Response to Reviewer 1 comment No. 4: As suggested by the reviewer, throughout the revised manuscript, the term “reaction rate” was replaced by the term “kinetic constant”.

5.      The authors state that the data provided is the average of three experiments, but they do not comment the associated report.

Response to Reviewer 1 comment No. 5: In the revised manuscript the error bars depicting the standard deviation of PCT, TOC and H2O2 concentration values have been added to the plots (Fig. 2 a), b), c)), and the sentence referring to the data provided as the average of three experiments, has been substituted by a footnote in Fig.2.

6.      Please, adjust the significant figures in the kinetic constant values for Fe3+ formation from Fe2+.

Response to Reviewer 1 comment No. 6: In the revised manuscript, all the significant figures in the kinetic constant values were adjusted.

7.      The discussion is too dense. The conditions of the experiments are repeatedly exposed during the discussion because the authors have not selected a proper terminology (acronyms), which does not allow a straight-forward interpretation.

Response to Reviewer 1 comment No. 7: In the revised manuscript the discussion was alleviated and the acronyms for the experiments were changed and were used throughout the text in order to facilitate the interpretation of the results.

8.      By-products formed (BP1 & BP2) must be inexcusably identified otherwise any further discussion has nil interest.

Response to Reviewer 1 comment No. 8: The by-products have been identified as benzoquinone and hydroquinone. In the revised manuscript this point has been clarified and discussed in lines 400-428.

9.      Since these by-products are unknown, the subsequent toxicity analysis loses its significance since any effect cannot be assigned to specific compounds (at least families of compounds). In addition, the toxicity study, of course well conducted, is not completely relevant as these effluents are typically derived back to the sewage biological plants, so it is more relevant the remnant toxicity against microorganism in the biological sludge.

Response to Reviewer 1 comment No. 9:

In the revised manuscript, as commented in the previous answer, the by-products have been identified as benzoquinone and hydroquinone.

Regarding the other comment about the selection of cytotoxicity tests, it was decided to use cell lines culture because we were not addressing a specific case study, for which the composition of the biological sludge is known. Hence, with the purpose of addressing a more generalizable case study, it was decided to select a system with an even higher sensitivity than the conventional one based on the use of specific bacteria (e.g.: vibrio fishery). This point has been clarified in the revised manuscript (lines 261-264).

10.   Conclusions are too long, too.

Response to Reviewer 1 comment No. 10: In the revised manuscript, the conclusion were shortened.

Reviewer 2 Report

Article Removal of Paracetamol Using Effective Advanced Oxidation Processes written by Francesca Audino, Jorge Mario Toro Santamaria, Luis Javier Del Valle Mendoza, Moisès Graells Sobre and Montserrat Pérez-Moya and submitted to International Journal of Enviromnantal Research and Public Health as a draft no. 408871 deals with the important issue pharmaceuticals decomposition.

The Authors applied 3 treatment processes for paracetamol removal from water. The article is interesting and could be considered for publication.

On the other hand, while reading I found some information missing, not clear or confusing. Below I enclose list of my comments:

Abstract – no information about kinetics, toxicity, reagent doses and efficiencies are given. It is only information what was done, instead of what were the results. Abstract has to be rewritten.

Literature review dealing with PCT is scarce -  only few articles dealing with PCT or pharmaceuticals in general. Deeper literature review should be done.

Only aqueous PCT solution was used. As the study “is aimed at providing evidence for deciding the extent of the AOP treatment before proceeding to a conventional biological treatment” (lines 99 – 101), why the authors do not used wastewater (for at least optimal samples)?

pH range (line 204 and others) 2.8 +/-0.2 (range 2.6 – 3.0). Why it was not kept constant? It could affect results.

Lines 274 – 290 it seems to be in Materials and Methods chapter.

How Fenton/photo-Fenton was terminated? It could strongly affect iron forms and hydrogen peroxide concentration?

Author Response

First of all the authors would like to thank the reviewers for the effort spent in improving with all their comments and suggestions the quality of the manuscript.

Below you can find the answers to each of the comments and questions arisen.

Reviewer #2:

1.      Abstract – no information about kinetics, toxicity, reagent doses and efficiencies are given. It is only information what was done, instead of what were the results. Abstract has to be rewritten.

Response to Reviewer 2 comment No. 1: The abstract was rewritten according to the reviewer’s instructions.

2.      Literature review dealing with PCT is scarce -  only few articles dealing with PCT or pharmaceuticals in general. Deeper literature review should be done.

Response to Reviewer 2 comment No. 2: The introduction was improved by including 13 relevant references dealing with the selected model contaminant (lines 96-99) as well as with other pharmaceuticals (lines 37-40).

3.      Only aqueous PCT solution was used. As the study “is aimed at providing evidence for deciding the extent of the AOP treatment before proceeding to a conventional biological treatment” (lines 99 – 101), why the authors do not used wastewater (for at least optimal samples)?

Response to Reviewer 2 comment No. 3: As clarified in the reviewed version of the manuscript (lines 110-113): “distilled water was set as water matrix in order to study the pure and specific degradation of paracetamol and of its by-products as well as their effect on the toxicity evolution and so avoiding the interference of other organic substances that are present in a real wastewater matrix with an uncertain composition”.

The previous sentence: “the study is aimed at providing evidence for deciding the extent of the AOP treatment before proceeding to a conventional biological treatment”  was certainly misleading and for this reason the aim of the work has been rewritten (see lines 84-94). Of course, we totally agree with the reviewer and the next step will surely be to investigate the influence of the water matrix.

4.      pH range (line 204 and others) 2.8 +/-0.2 (range 2.6 – 3.0). Why it was not kept constant? It could affect results.

Response to Reviewer 2 comment No. 4: No specific pH control method was used. Instead, pH was monitored during all the experiments using a pH sensor connected to a SCADA system (this has been clarified in lines 202-205 and line 237 of the revised manuscript) .

The following Fig. A, shows the pH evolution in the case of Fenton and photo-Fenton experiments under low and high Fenton reagents loads and in case of the photo-Fenton experiment performed dosing the H2O2 dosage experiments.

As can be observed, pH is between the range described by Pignatello et al. (1992, 2006) as the optimal one. According to Pignatello et al. (2006) : “care has to be taken to keep the pH in a range (2.5 < pH < 3–4) where the Fe(OH)2+ species exists in appreciable concentration and the bulk of the iron remains soluble”.

As can be seen in Fig. A, all experiments can be guaranteed to be between this range. This is also confirmed by the total iron evolution showed in the manuscript in Fig. 1 a) and b).

The sentence in lines (340-341) has been extended and improved in order to clarify this issue in regard of Ph.

The references to the works by Pignatello are also included in the manuscript:

28.   Pignatello J.J.; Oliveros, E.; MacKay, A. Advanced Oxidation Processes for Organic Contaminant Destruction Based on Fenton Reaction and Related Chemistry. Critical Reviews in Environmental Science and Technology, 2006, 36: 1-84. http://dx.doi.org/10.1080/10643380500326564

29.   Pignatello, J.J. Dark and photoassisted Fe3+ — catalysed degradation of chlorophenoxy herbicides by hydrogen peroxide.Environ. Sci. Technol., 1992, 26 (5), pp 944–951, https://doi.org/10.1021/es00029a012

Fig. A. pH   evolution during Fenton and photo-Fenton experiments performed under low   Fenton reagent loads (FENTON _LOW and PHOTO- FENTON _LOW, respectively) and   high Fenton reagent loads (FENTON_HIGH and PHOTO- FENTON _HIGH, respectively)   as well as for the photo-Fenton experiment performed by dosing H2O2   (PHOTO- FENTON _DOSAGE)

5.      Lines 274 – 290 it seems to be in Materials and Methods chapter.

Response to Reviewer 2 comment No. 5: As suggested by the reviewer, the description of the design of experiments (lines 274 – 290 in the old version of the manuscript) has been moved to the Materials and Methods section of the revised version of the manuscript (lines 148-169).

Besides, the materials and methods section has been rewritten and shortened as suggested by the other reviewer.

6.      How Fenton/photo-Fenton was terminated? It could strongly affect iron forms and hydrogen peroxide concentration?

Response to Reviewer 2 comment No. 6:

As clarified in the revised manuscript (lines 223-226), the standard methods that have been strictly followed for the measurements of H2O2 (Nogueira et al. 2005) and iron species (ISO 6332), ensure the proper determination of the oxidant and catalyst concentrations during Fenton and photo-Fenton experiments.

Particularly, the spectrophotometric technique described by Nogueira et al. (Nogueira, R.F.P.; Oliveira, M.C.; Paterlini, W.C. Simple and fast spectrophotometric determination of H2O2 in photo-Fenton reactions using metavanadate. Talanta, 2005, 66(1):86–91, https://doi.org/10.1016/j.talanta.2004.10.001) was followed for the measurement of hydrogen peroxide concentration. The method is based on the reaction of hydrogen peroxide with metavanadate ammonium (the latter is added in excess so to ensure that all hydrogen peroxide reacts), in an acid medium, which causes the formation of the peroxovanadium cation (VO23+) that is a red-orange color complex with a maximum absorbance at 450 nm.

Conversely, the iron species were analysed using the 1, 10-phentranoline method following ISO 6332 based on the absorbance measurements of the Fe2+-phenantroline complex at 510 nm.

Total iron concentration ( ), is measured by the means of ascorbic acid that can convert all ferric ions ( ) to ferrous ions ( ). Then, for difference, ferric ion concentration can be determined (  = - ).

This method is based on the principle that dissolved ferrous iron forms a stable orange-red colored chelate complex with 1,10- phenanthroline, which does not change color between pH 3 and 9 and has a molar extinction coefficient of 11720 ± 60 L mol-1 cm-1 at 510 nm. Although this range is large enough to ensure the quantitative formation of the desired complex, the pH, during the measurements, is maintained between 3 and 3.5 by using a buffer solution in order to ensure a rapid quantitative development of the color. The presence in the solution of H2O2 interferes with the above described method, as it oxidizes Fe2+ to Fe3+ which does not form complexes with phenanthroline. The interference caused by H2O2 is avoided by using ascorbic acid which reduces any Fe3+ to Fe2+ and destroys H2O2 so that the iron measured is the total iron present in the dissolved form in the sample.

Round  2

Reviewer 1 Report

Please find attached the comments for the revised version of the manuscript, which is almost ready to be accepted.

Author Response

The authors would like to thank the reviewer 1 for the comments and suggestions that were really helpful to improve the present work.

Below you can find the response (highlighted in blue) to the questions and comments arisen by the reviewer 1.

Please note that all the new changes are highlighted in green in the reviewed version of the manuscript.

1.      I suggest conducting a later revision of the manuscript by an English-speaking native. At the bottom of the comments, you can find some collected mistakes, but there are sure others inadvertent during the reading.

The manuscript was carefully and thoroughly revised.

2.      The authors comment that distilled water was used for the preparation of solutions, which is not very frequent. Do you mean distilled or deionised water? If it is the former, then report the equipment to produce it and its quality.

The reviewer is right, deionised water was used as water matrix. We used distilled water to perform some preliminary experiments in a lab scale pilot plant (that are not included in the present manuscript) and we did not realize that we were erroneously referring to distilled water.  Hence, many thanks to highlight such a mistake.

In the revised manuscript “distilled water” has been replaced by “deionised water” and the reference of the provider of the used deionised water has been also added to the reagents and chemicals section (page 4 lines 138-140).

3.      Virr is first used on page 4 but only defined in page 6. Please, correct this misconnection.

The definition of VIRR is now given in page 4 (line 156-158) where it is first used as properly observed by the reviewer.

4.      Please, include the recirculation flow rate applied in each case in order to assess the batch operation assumption.

The recirculation flow-rate used to ensure perfect mixing during Fenton and photo-Fenton assays has been added (page 6, line 237-238).

5.      In the TOC analysis protocol, the samples were refrigerated to slow down the degradation outside the reactor. Please, provide some assessment on how efficient is this method.

A preliminary experiment was performed in order to test the efficiency of this method. Particularly, during this experiment, a sample was taken after 5 min of reaction, and immediately after the sample was taken, the TOC concentration was measured by maintaining the sample refrigerated. Then, the TOC concentration of this same sample was measured 5 times more (always under refrigeration) and each measurement was performed after 1 hour from the previous one. By analyzing the TOC concentrations values that were obtained it was possible to observe that the measurement error was inside the error range of the equipment (±2 mg L-1).

6.      In the subsection 2.3, why do you refer to the kinetic model as semi-empirical? Where is the empirical part?

The term “semi-empirical model” has been used to refer to a balanced approach between detailed first-principles modeling and pure empirical modeling (response surface).

The semi-empirical model proposed by Pérez-Moya et al. 2011 on one hand describes the degradation of the model contaminant by measuring a lumped parameter such as the concentration of total organic carbon, [TOC]. On the other hand, the model is intended to overcome the limitations of pure statistical modeling and has been expressed in terms of physically meaningful factors that may be used for quantitatively assessing the performance of the process.

This concept has been clarified in the revised version of the manuscript (page 7 line 294-299).

7.      In my opinion, the cytotoxicity assay applied has not been a good selection since hardly can be used to extrapolate the impact on conventional sewage sludge plants or even to compare against other standard tests. For this goal, respirometry-based assays are much better as they provide assessment for real plants. I recommend moving to other tests for future works in this field.

We will surely take into account your suggestion for future works.

8.      Corrections:

- Page 2, line 46, remove of after and .

- Page 2, line 58, remove the existing comma.

- Page 2, line 60, replace showed with shown. Again in page 16, line 490.

- Page 2, lines 77-78  appears twice in the list of species.

- Page 6, line 246-247, replace from 4.2 and 0.8 L min-1 with from 4.2 to 0.8 L min-1 or

between 4.2 and 0.8 L min-1 .

- Page 6, line 248, the units for the effective absorbing path should be cm, not cm2.

- Repeatedly, kinetics and species apply to both singular and plural forms. Please, check throughout the text.

- Page 12, line 363, substitute 6 percentage point improvement for 6% more .

- Along the manuscript, the authors use  for denoting hydroxyl radicals. Please, use a single form, preferably the latter.

- Page 16, line 496, change From one side to On the one hand . Similarly (page 16, line 506), change From the other side to On the other hand.

All suggested corrections have been made in the reviewed manuscript.

Reviewer 2 Report

My comments from previous review have been applied. I do not have additional ones. I suggest to accept this article in its present form.

Author Response

Thank you for your valuable comments and suggestion for our manuscript.